# Application of Human Stem Cells to Model Genetic Sensorineural Hearing Loss and Meniere Disease

**DOI:** 10.3390/cells12070988

**Published:** 2023-03-23

**Authors:** Mar Lamolda, Lidia Frejo, Alvaro Gallego-Martinez, Jose A. Lopez-Escamez

**Affiliations:** 1Otology and Neurotology Group CTS495, Department of Genomic Medicine, GENYO-Centre for Genomics and Oncological Research-Pfizer, University of Granada, Junta de Andalucía, PTS, 18016 Granada, Spain; 2Division of Otolaryngology, Department of Surgery, Instituto de Investigación Biosanitaria, ibs.GRANADA, Universidad de Granada, 18071 Granada, Spain; 3Sensorineural Pathology Programme, Centro de Investigación Biomédica en Red en Enfermedades Raras, CIBERER, 28029 Madrid, Spain; 4Meniere’s Disease Neuroscience Research Program, Faculty of Medicine & Health, School of Medical Sciences, The Kolling Institute, University of Sydney, Sydney, NSW 2006, Australia

**Keywords:** human-induced pluripotent stem cell, inner ear disorders, disease modeling, sensorineural hearing loss, Meniere disease, biomedical applications

## Abstract

Genetic sensorineural hearing loss and Meniere disease have been associated with rare variations in the coding and non-coding region of the human genome. Most of these variants were classified as likely pathogenic or variants of unknown significance and require functional validation in cellular or animal models. Given the difficulties to obtain human samples and the raising concerns about animal experimentation, human-induced pluripotent stem cells emerged as cellular models to investigate the interaction of genetic and environmental factors in the pathogenesis of inner ear disorders. The generation of human sensory epithelia and neuron-like cells carrying the variants of interest may facilitate a better understanding of their role during differentiation. These cellular models will allow us to explore new strategies for restoring hearing and vestibular sensory epithelia as well as neurons. This review summarized the use of human-induced pluripotent stem cells in sensorineural hearing loss and Meniere disease and proposed some strategies for its application in clinical practice.

## 1. Introduction

Stem cell culture is essential in biomedical research to study cellular and developmental biology [1,2]. Several biological phenomena specific to humans, such as brain development and inner ear formation amongst others, are not reproduced in animal models. This limitation promoted the generation of cellular models to reproduce more exactly human biology and development [1], leading to a better understanding of genetic disorders.

Stem cell science presents a highly promising direction in translational medicine, allowing the proliferation and differentiation of any cell type. Stem cell models are more clinically appropriate to study the molecular pathophysiology of certain diseases, and further develop better therapeutic strategies [2]. Therefore, human pluripotent stem cells (hPSCs) are progressively used to model rare diseases, as an alternative to animal models which may raise research costs and ethical issues [1]. Disease modeling of human disorders may allow the development of new therapies for rare diseases [1]. Henceforth, human organoids provide a unique opportunity to study them in a multicellular environment and complement animal models [3].

The most common sensory disorder is hearing loss, affecting around 15% of the population [4]. Sensorineural hearing loss (SNHL) and vestibular disorders are caused by damage to the sensory epithelia and neurons, which do not regenerate in humans. These could originate from a variety of causes, such as genetic and environmental factors, aging, and ototoxic drugs [5].

Extensive studies using animal models expanded our knowledge of the human inner ear function and disease, although access to the inner ear in mammalian animal models is limited, slowing research progress in the field [6]. Access to the human inner ear is also strictly restricted since tissue sampling may lead to irreversible damage and profound hearing loss or vestibular impairment. In addition, non-invasive clinical imaging techniques such as computed tomography or MRI, do not provide sufficient resolution to investigate most pathologies of the inner ear at the cellular and molecular levels [7].

Researchers successfully derived inner ear progenitors and sensory cells from hPSCs [8,9,10,11] using different induction protocols to form hair cells-like cells. Many of these protocols were carried out entirely in 2D culture providing a homogeneous cell population [12,13,14]. In contrast, 3D organoid systems containing multiple cell types can better recapitulate in vivo composition of the organ [15,16,17,18]. Since hair cells (HCs), supporting cells (SCs), and their associated neurons are damaged in SNHL, gene and cell-based therapies may restore hearing and vestibular impairment.

This scoping review describes the major achievements, gaps in research, and challenges to translating the application of hPSCs into clinical practice. Furthermore, we present their application in monogenic deafness and the potential use to model Meniere disease (MD).

## 2. Generation of Human Pluripotent Stem Cells

Human pluripotent stem cells (hPSCs), including human embryonic stem cells (hESCs) and hiPSCs, emerged as a new model system that offers unique advantages for disease modeling [2].

hESCs are derived from the inner cell mass of the blastocyst of an embryo, while hiPSCs are derived from adult somatic tissues. Since their discovery in 2007 [19], hiPSCs emerged as an alternative to hESCs, because they can be obtained from individual patients or healthy donors and immune rejection can be avoided when they are transplanted autologously [20,21]. Since then, researchers showed the potential of reprogramming to convert a given somatic cell type to an iPSC state.

Reprogramming methods may be divided into two categories: integrative and non-integrative (Table 1). Integrative reprogramming approaches generate heterogeneous hiPSCs lines and present genomic mutation risks, which could obscure comparative analysis between lines. So, to avoid the risk of tumorigenesis, non-integrative reprogramming methods were designed, including non-integrating viral vectors such as Sendai virus (SeV), adenovirus, and non-viral vectors such as episomal DNA, Piggy-Bac transposons, modified synthetic mRNA (modRNA), microRNA, and recombinant proteins.

The most commonly used reprogramming methods to generate hiPSCs are SeV, synthetic mRNAs, and episomal vectors, due to their high reprogramming efficiency and their wide application to different cell types [20,22,23,24,25,50,51,52,53]. In addition, SeV reprogramming is highly effective, with a lower workload and no non-appearance of viral sequences in most cell lines at higher passages in culture. Moreover, the combination of bFGF, ascorbic acid (AA), and Y-27632 dihydrochloride-ROCK inhibitor in the culture medium increase the reprogramming efficiency [54,55,56,57] (Figure 1). However, in recent years, modRNA-based somatic reprogramming became a more effective alternative to the SeV system, since the modRNA molecules encode multiple pluripotent factors and can be applied successfully in reprogramming somatic cells [24,25,29].

Selecting the most suitable reprogramming method and implementing good manufacturing procedures for hiPSCs generation is key to obtaining high-quality hiPSCs cell lines.

## 3. Disease Modeling by Human-Induced Pluripotent Stem Cells

The advantages of hiPSCs are well-known at many distinct levels such as their capacity to self-renew indefinitely in culture and differentiate to any cell type in the human body. So, hiPSCs are the ideal source for generating a patient-derived personalized disease model [1,2].

Experimental human modeling of inherited rare disorders provides insight into the cellular and molecular mechanisms involved in the disease. It is worth mentioning the early success history of disease modeling using hiPSCs, which was based on neurodegenerative diseases [58]. Neuronal differentiation was one of the initially differentiated target tissues from several sources of stem cells due to their potential future therapeutic usage [59,60,61]. Neurodegenerative disorders such as amyotrophic lateral sclerosis, Huntington’s disease, Parkinson’s disease, and Alzheimer’s disease were among the most studied [62,63,64].

Recent advances in organoid technology revolutionized the in vitro culture tools for biomedical research by creating powerful 3D models to recapitulate the cellular heterogeneity, structure, and functions of the primary tissues, compared with 2D cell models. Hence, it is a methodology with many translational applications such as regenerative medicine, drug discovery, and precision medicine. In 2011, the first organoids from hPSCs were generated to recreate intestinal tissue in vitro [65]. In 2012, the first retinal organoids were generated from hPSCs [66]. These human retinal organoids are larger than mouse organoids and can grow into multilayered tissue containing both rods and cones [66,67]. In 2013, Lancaster et al. further established the 3D cerebral organoids containing different brain regions within single organoids [68].

## 4. Clinical Applications in Genetic Deafness and Vestibular Disorders

According to the World Health Organization (WHO), hearing impairment, the partial or total inability to hear sounds, is among the top 10 disabilities of today’s society. (https://www.who.int/newsroom/factsheets/detail/deafness-and-hearing-loss, accessed on 17 November 2022). 

Congenital SNHL is estimated to have a genetic origin in 70% of cases and, mutations can affect the organ of Corti, the spiral ganglion, and almost any part of the auditory pathway [41,42]. Monogenic SNHL are considered rare disorders [43,44]. There are around 126 genes associated with non-syndromic SNHL (https://hereditaryhearingloss.org/dominant, accessed on 10 November 2022) and many of them are related to inner ear homeostasis and mechano-electrical transduction [69,70,71,72].

Several otologic conditions may show fluctuating SNHL, including Meniere Disease (MD), a debilitating condition, characterized by episodes of spontaneous vertigo, tinnitus, and aural fullness [73,74]. It is associated with the progressive accumulation of endolymph in the cochlear duct [75]. The diagnosis is based on the characteristic presentation of the different clinical symptoms mentioned during the vertigo attacks [76]. MD prevalence is higher in European than Asian and African populations with a range of 10–225 cases/100,000 individuals. Most patients suffer from SNHL in one ear, but 25–40% of these patients with unilateral SNHL may develop hearing loss in the contralateral ear after several years [77]. MD is known to have a genetic predisposition [74] and familial MD is found in around 8–10% of cases with several genes involved [78,79,80,81,82]. Some forms of monogenic SNHL and familial MD are rare diseases which may benefit from gene and cellular therapy.

Despite the extensive occurrence of genetic SNHL in the world, there are no Food and Drug Administration (FDA)-approved cellular or molecular therapies [4,50]. Current treatments for human SNHL and MD are medical therapy using steroids, hearing aids, surgery to correct the cause of the hearing loss, or cochlear implants [83,84,85,86,87,88,89]. Though these devices offer significant relief of the moderate and severe SNHL by amplifying sound or directly electrically stimulating the auditory nerve, they have significant limitations in terms of speech discrimination in complex acoustic environments [90]. These medical devices require the presence of functional auditory neurons in the inner ear. Therefore, in recent years, new studies focused on possibilities for neuronal replacement, including exogenous stem cell transplantation and endogenous cell source replacement. Several studies proved that neural stem cell transplantation in the inner ear has an important therapeutic effect on the activation and regeneration of cells, restoring damaged neurons [91,92,93]. However, more research is still needed to improve and standardize the protocol for differentiating stem cells into inner ear HCs and neurons.

Transcriptional networks are key in governing the regeneration or replacement of auditory neurons from stem cells. Development of the inner ear is an organized molecular transformation of a set of epidermal cells (the otic placode) into the fully developed ear with its neurosensory component, necessary for signal extraction and transmission, and the non-sensory component, forming the labyrinth necessary for directing sensory stimuli to specific sensory epithelia [6,94]. The main genes involved in neurosensory development in the inner ear are *MYO7A*, *HES5*, *SOX2, NEUROG1, NEUROD1,* and *POU4F1* [80,95,96,97,98].

Since animal models are only able to represent the chronic end stages of the disease when permanent damage of sensory epithelia occurred, understanding, and identifying the transcription factors involved in the development and survival of auditory neurons is vital for the generation of disease models and the identification of more effective treatments for hearing loss in the future.

### 4.1. Modeling Inner Ear Disorders: 2D and 3D Cell Culture

The inner ear is highly complex; both the anterior and posterior labyrinth are connected, and the extent of involvement in each organ may vary, resulting in hearing or vestibular disorders. Consequently, inner ear disorders may be caused by damage to sensory epithelia and neurons, which do not regenerate to any clinically relevant extent in humans. Since the pathophysiology of certain types of SNHL and MD have not yet been explained at cellular and molecular levels, it is difficult to generate an animal model that accurately reflects these diseases.

Nevertheless, extensive studies used animal models, including frog, zebrafish, chick, and several species of mammals that expanded our knowledge of the human inner ear function and disease, although access to the inner ear in mammalian animal models is limited, slowing progress in the field. Additionally, these animal models are only able to represent the chronic end stages of disease with permanent loss of hearing and vestibular function [5].

Access to the human inner ear is also strictly restricted since tissue sampling is challenging and leads to irreversible damage. In addition, non-invasive clinical imaging techniques, such as computed tomography or MRI, do not provide enough resolution to investigate most pathologies of the inner ear at the cellular and molecular levels [5,7].

For this reason, the best option to study the inner ear development and disease is to generate human cell models. Several protocols were devised to direct hPSCs into inner ear HCs and neuron-like cells [99,100,101,102]. The efficiency, reproducibility, and scalability of these protocols were enhanced by incorporating knowledge of inner ear development [13,17,18,67,68,69,70]. Early studies on the transplantation of hPSCs-derived otic progenitors were successful in certain animal models [103], but the hearing was transiently restored, and long-term cell survival continues to be a major challenge. Understanding the complex sequence of transcriptional changes and signaling pathways in vivo in inner ear development is critical to the successful differentiation of hPSCs into inner ear tissues, such as HCs, SCs, and neurons in vitro.

Recent years saw a surge in the number of studies that were conducted in vitro 2D and 3D hiPSC models to study auditory and vestibular disorders.

#### 4.1.1. Sensory Epithelia

Several induction protocols were developed to differentiate hPSCs into HC-like cells. Many of these protocols start with the generation of floating embryoid bodies (EBs) followed by the combination of small molecules and/or recombinant proteins that become adhesive 2D cell cultures after approximately 5 days and other protocols were carried out entirely in 2D, which can provide a homogeneous cell population [12,13,14,67,72]. In contrast, 3D organoid systems contain multiple cell types and more exactly recapitulate in vivo composition of an organ.

The initial steps of inner ear development require the formation of the ectodermal germ layer followed by the generation of the pre-placodal ectoderm (PPE). Protocols first inhibit transforming growth factor β and WNT signaling and activate BMP to promote non-neural ectoderm (NNE) development, while reducing mesoderm development [13,17,72]. Insulin-like growth factor 1 promotes the fate of the anterior ectoderm, where the PPE emerges [104]. The PPE gives rise to most of the cranial placodes, including the otic placode. Physical environmental signals provided by extracellular matrices, such as Matrigel, improve the efficiency of differentiation from hPSCs, as well as the resulting cellular assemblage [15,17,18]. The addition of Matrigel in the 3D inner ear organoid systems facilitates the formation of fluid-filled vesicles containing HCs and SC-like cells [17]. However, vestibular tissue-like organoids derived from hPSCs using the rotary cell culture system form HC-like cells on the surface of the organoids [18]. To date, hPSCs-derived HC-like cells display molecular markers and electrophysiological properties of vestibular HCs, not cochlear HCs. The discovery of alternative small molecules or culture conditions to improve the generation of cochlear HCs from hPSCs is still needed.

SCs play an active role in ion metabolism necessary for HCs’ function. The connexin proteins are the most abundant gap junction proteins expressed in the SCs. Mutations in the *GJB2* gene, encoding connexin 26 (CX26), are the most common cause of autosomal recessive non-syndromic SNHL [105]. An in vitro model for the homozygous 235delC mutation in *GJB2* was developed from hPSC to develop a therapy for deafness [106] (Figure 2).

Frejo et al. generated a hiPSC line derived from an MD patient. This model was differentiated into HCs by a 2D protocol based on Boddy et al. [101]. This method consisted of the generation of otic epithelial progenitors (OEPs) and, consequently, differentiation to HCs-like cells (Figure 3).

Moreover, we started to generate inner ear organoids from the hiPSC-MD model derived from a patient with mutations in *DTNA* and *FAM136A* genes (https://hpscreg.eu/cell-line/GENYOi007-A, accessed on 15 March 2023) for studying the development of the inner ear in this patient. This model could explain how these mutations found in *DTNA* and *FAM136A* genes in a Spanish family with three affected women in three consecutive generations (autosomal dominant inheritance pattern) affect the development and functionality of the system itself when sensory organs mature [75,76,82,107,108]. Koehler et al. described the protocol [17] to generate inner ear sensory epithelium harboring HCs using an in vitro 3D differentiation system from hiPSCs. Cells were treated with recombinant proteins that modulate BMP, FGF, and WNT signaling pathways to induce the sequential formation of NNE, otic-epibranchial progenitor domain (OEPD), and otic placodes. The otic placodes subsequently underwent self-guided morphogenesis to form inner ear HCs and SCs (Figure 4).

#### 4.1.2. Sensory Neuron

Early neural induction protocols from PSCs using stromal cells derived from skull bone marrow, resulted in efficient dopaminergic neuron production [110]. These protocols were modified to derive inner ear sensory neurons and glial cells from PSCs. However, many of the available protocols were tailored to generate hPSCs-dopaminergic neurons due to their role in neurodegenerative disorders, such as Parkinson’s disease [111]. In contrast, glutamate was the main neurotransmitter for the synaptic transmission between HCs and afferent sensory neurons within the inner ear. So, the derivation of glutamatergic neurons was a key stage for recapitulating afferent neural transmission in the inner ear [112].

As with the sensory epithelia, inner ear sensory neurons are derived from the otic placode. Therefore, some of the early induction steps for hPSCs-derived sensory neurons are based on known developmental pathways of the otic placode. Small molecules, such as FGFs, BMP, SHH, and noggin, are used to support neuronal outgrowth from the otic placode [113,114,115]. POU4F1 and β-III tubulin are commonly used to verify neuron formation, with glutamate receptors and transporters subsequently used to confirm subtype-specific derivatives. Matsuoka et al. showed that ~90% of neuron-like cells were peripherin^+^ and β-III tubulin^+^, but only ~46% were POU4F1^+^ [114]. To recapitulate the peripheral neural circuit in vitro, the model must contain both sensory epithelia and innervating neurons.

Two systems potentially can generate both tissues simultaneously, providing complementary research tools for disease modeling: 2D and 3D cell cultures (Figure 5).

The treatment of hESCs with FGF3 and FGF10 for 10–12 days in 2D culture gives rise to both otic epithelial progenitors (OEPs) and otic neural progenitors (ONPs), which can be distinguished based on their morphology [103]. HC-like cells can be derived from OEP after inhibiting Notch and supplementing with RA and epidermal growth factor (EGF) [67,71]. Sensory neuron-like cells can be derived from ONPs using bFGF and SHH followed by BDNF and NT3 supplementation [103]. Co-culture of these derived HCs and sensory neurons form neural connections in vitro. However, this method required separate induction protocols before co-culturing them [116].

The 3D inner ear organoids contain sensory epithelia and sensory neuron-like cells [15,16,17]. Furthermore, several studies replicated these protocols and used the systems for different applications, such as electrophysiological studies and disease modeling [117]. The inner ear organoid system is a powerful tool to study peripheral sensory neural networks in the inner ear in vitro. Human inner ear tissue derived from hPSCs also offers the chance to explore developmental biology and understand the differences between mice and human inner ear development. Moreover, it would enable both in vitro screening of drug candidates for the treatment of hearing loss and balance dysfunction and a source of cells for cell-based therapies of the inner ear. Nie and Hashino described a 3D protocol to form inner ear neurons from hPSC [109]. Other studies generated otic organoids with neuron-like cells from hiPSC models combining the 2D and 3D systems in the same differentiation protocol [100].

Stem cell science was also applied to generate a hiPSCs model obtained from patients with severe tinnitus and rare variants in the *ANK2* gene, and differentiating them to inner ear neurons using a 2D system [101]. This protocol consists of two phases, a first phase in which hiPSC-derived otic neural progenitors (ONPs) are generated by inhibiting Wnt signaling, which is accompanied by subsequent activation of this pathway, and a second phase consisting of ONPs expansion and enrichment, which will later differentiate into inner ear neurons (Figure 6).

## 5. Clinical Trials in Inner Ear Disorders

Several clinical trials were performed using human stem cells to treat SNHL. Autologous human umbilical cord blood stem cells were used for over twenty years, with excellent safety records. A study in 2015 used umbilical cord blood stem cells to improve inner ear function, audition, and language in children [118]. The potential of human mesenchymal stem cells was evaluated for years for regenerating inner ear HCs [119].

The major achievements in disease modeling using a hiPSC-derived inner ear for genetic SNHL include the genes *MYO7A, MYO15,* and *MERRF* in HCs [70,89,90], and *GJB2* and *SLC26A4* (Pendred syndrome) in SCs [106,120,121,122,123,124] (Table 2). These studies defined the molecular mechanisms involving each gene and showed the cellular effects of each mutation. Recently, a clinical trial using iPSC derived from patients with Pendred syndrome to generate cochlear cells allowed the study of the effects of low-dose oral administration of sirolimus for fluctuating and progressive hearing loss [125].

However, many research gaps were not addressed and more than 150 genes in SNHL and familial MD remain to be modeled (https://deafnessvariationdatabase.org/, accessed on 4 December 2022). Some of the current limitations of the use of hPSCs to investigate inner ear disorders include laborious culture protocols that are sometimes time-consuming and not reproducible, variable efficiency of tissue derivation, and limited differentiation and integration in host tissues. Deciphering the disease molecular mechanisms is the first step to finding a drug target that may offer new opportunities for therapy.

## 6. Conclusions

Human iPSCs are a reliable model to study the functional consequences of rare variants in early stages of inner ear disease development and design novel gene therapies to restore the phenotype. The use of 3D models to generate patient-specific organoids allows us to elucidate the effect of rare variants in genetic deafness and vestibular disorders, investigate the molecular mechanisms underlying the disease, and analyze the effects of genetic mutations on the phenotype at the cellular and organoid levels. Modeling inner ear diseases by differentiating hiPSCs is a promising tool for designing novel therapies for the treatment of SNHL and MD.

## Figures and Tables

**Figure 1 cells-12-00988-f001:**
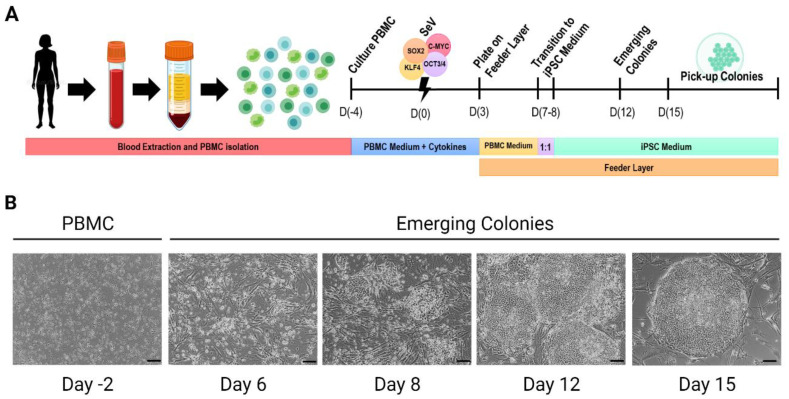
Cell reprogramming of human PBMC. (**A**). Generation of hiPSC lines by SeV cell reprogramming technology. (**B**). Images of cell reprogramming procedure. Scale bar = 100 μm. (Created with BioRender.com, accessed on 1 December 2022).

**Figure 2 cells-12-00988-f002:**
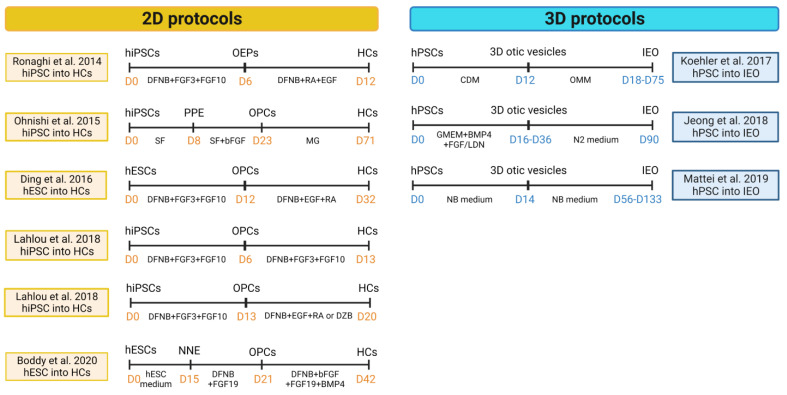
2D and 3D hPSCs-derived inner ear hair cells protocols [12,13,14,15,17,18,101,104]. (Created with BioRender.com, accessed on 22 March 2023).

**Figure 3 cells-12-00988-f003:**
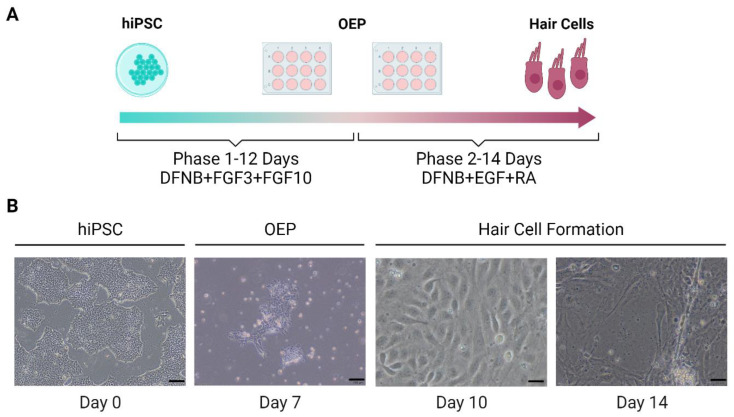
Schematic representation of 2D differentiation protocol of hPSCs-derived hair cells. (**A**). Protocol of the generation of inner ear hair cells from hPSCs. (**B**). Images obtained by microscopy of hair cell formation. Scale bar at day 0 and 7 =100 μm and at day 10 and 14 =20 μm. OEP: Otic epithelial progenitor. (Created with BioRender.com, accessed on 4 December 2022).

**Figure 4 cells-12-00988-f004:**
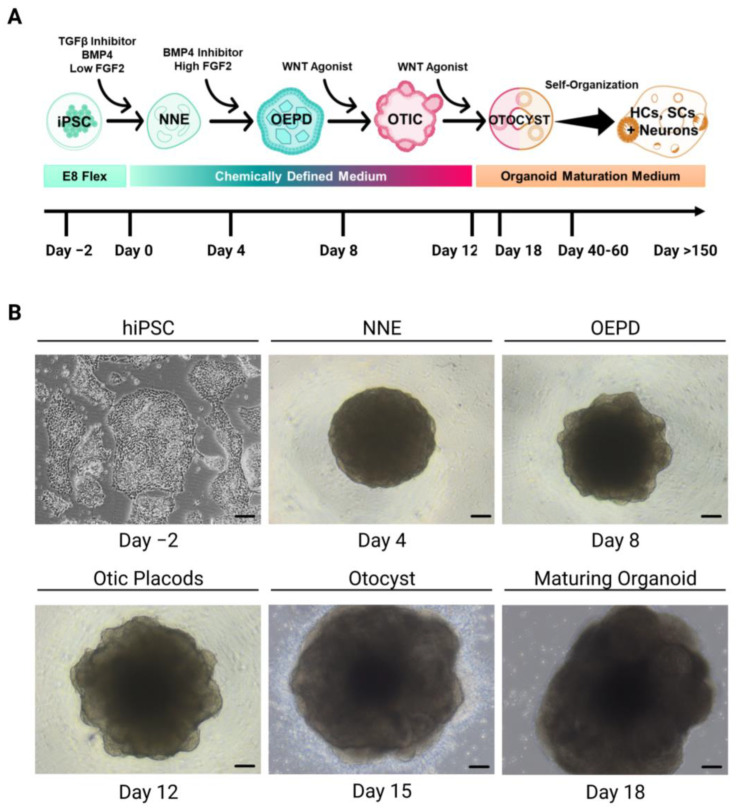
Generation of inner ear organoids from hPSCs (modified from Nie and Hashino 2020 [109]). (**A**). 3D differentiation protocol of hPSCs into inner ear organoids from Koehler et al. [17]. (**B**). Images of inner ear organoids generation. NNE: Non-neural ectoderm. OEDP: Otic-epibranchial progenitor domain. OTIC: Otic Placodes. (Created with BioRender.com, accessed on 22 March 2023).

**Figure 5 cells-12-00988-f005:**
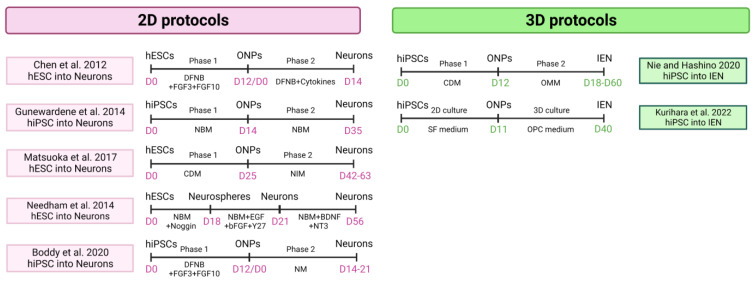
Cellular models of hPSC-derived 2D and 3D inner ear neurons protocols [100,101,103,109,113,114,115]. NBM: neurobasal medium. CDM: chemically defined medium. NIM: neural inducing medium. NM: neuralizing medium. (Created with BioRender.com, accessed on 22 March 2023).

**Figure 6 cells-12-00988-f006:**
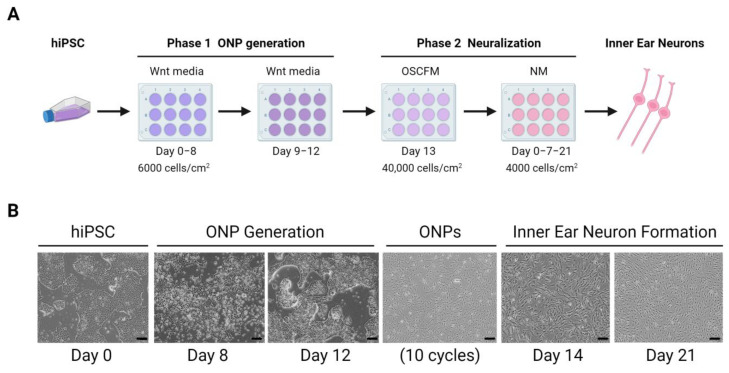
Inner ear neurons generation from hiPSCs. (**A**). Differentiation protocol of hPSC-derived inner ear neurons in 2D culture from Boddy et al. [101]. (**B**). Images of inner ear neurons culture derived from hiPSC. Scale bar = 100 μm. ONP: otic neural progenitor. OSCFM: otic stem cell full media. NM: neuralizing media. (Created with BioRender.com, accessed on 22 March 2023).

**Table 1 cells-12-00988-t001:** Reprogramming methods used in human somatic cells.

	Type of Vector	Integration	Factors	References
**Viral**	Retrovirus	Yes	OSKM	[19,22,23,24]
			OSK	[25,26]
			OK	[27]
			Hsa-miR-302	[28]
	Lentivirus	Yes	OSKM	[29]
			OSNL	[30,31]
			OSKMNL	[32]
			OSN	[33]
			O	[34]
	Bacteriophage	Yes	OSKM	[35]
	Adenovirus	No	OSKM	[36]
	Sendai Virus	No	OSKM	[37,38]
**Non-viral**	piggyBac	No	OSKM	[39,40]
	Plasmid	No	OSNL	[41]
	Episomal Vector	No	OSKMNL	[42,43]
			OSKM*L	[44]
	Minicircle vector	No	OSNL	[45]
	Protein	No	OSKM	[46]
	mRNA	No	OSNL	[47]
			OSKM (L)	[48]
	microRNA	No	miR-200c, 302 369-3p/5p	[49]

O (OCT3/4); S (SOX2); K (KLF4); M (C-MYC); M* (L-MYC); N (NANOG); L (LIN28).

**Table 2 cells-12-00988-t002:** Genetic mutations causing inner ear disorders modeled using hPSCs.

Gene	RefSeq	Variant	Type of Inheritance	Model	References
** *MYO7A* **	NM_000260.4	c.1184G > A c.4118C > T	Compound heterozygous autosomal recessive	Hair Cells	[120]
** *MYO15A* **	NM_016239.4	c.4642G > A c.8374G > A	Compound heterozygous autosomal recessive	Hair Cells	[102]
** *MERRF/MT-TK* **	NC_012920.1	c.8344A > G	Mitochondrial	Hair Cells	[121]
** *SLC26A4* **	NM_000441.2	c.439A > G c.1229C > T c.2168A > G	Autosomal recessive	Supporting Cells	[123,124]
** *GJB2* **	NM_004004.6	c.235DelC	Autosomal recessive/Digenic	Supporting Cells	[106,122]

## Data Availability

Not applicable.

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
