# Peer review of "Application of Human Stem Cells to Model Genetic Sensorineural Hearing Loss and Meniere Disease"

_cells, 2023, doi:10.3390/cells12070988_

Round 1

Reviewer 1 Report

 A good review article on stem cell therapy applications in hearing loss. I believe that it will guide the researchers who plan to study on this subject.

Author Response

Thank you for reviewing our manuscript entitled “Application of Human Stem Cells to Model Genetic Sensorineural Hearing Loss and Meniere Disease”. We appreciate your comments and your consideration for publishing our review article.

Reviewer 2 Report

1.-They should be careful in their lenguage and in the literature citations. They have written on page seven

"Moreover, we have started to generate inner ear organoids from ..........(75,76)", but none of the authors was included in these references. Therefore they must rewrite the sentence.

2.- Table 2. The nomenclature is not correct since the type of inheritance is not “compound heterozygous” . The type of inheritance could be autosomal recessive, autosomal dominant, X-linked etc.......Please correct it.

Author Response

Thank you for your time to review our contribution. 

Point 1: They should be careful in their language and in the literature citations. They have written on page seven "Moreover, we have started to generate inner ear organoids from ..........(75,76)", but none of the authors was included in these references. Therefore, they must rewrite the sentence.

Response 1: We have corrected the references.

Now page 7, line 249, references 107 and 108.

Point 2: Table 2. The nomenclature is not correct since the type of inheritance is not “compound heterozygous”. The type of inheritance could be autosomal recessive, autosomal dominant, X-linked etc.......Please correct it.

Response 2: We have modified Table 2

Reviewer 3 Report

Well written. Please provide details of MD and DTA / FAM146A organoid which is not published yet. Since the title includes MD disease modeling this data should be discusses in this manuscript. These should be additional review for the manuscript including the data.

Author Response

Point 1: Well written. Please provide details of MD and DTA / FAM146A organoid which is not published yet. Since the title includes MD disease modeling this data should be discusses in this manuscript. These should be additional review for the manuscript including the data.

Response 1: Thank you for your comment. We have added the link to the repository of MD-hiPSC cell line (page 7, line 247). As mentioned in the manuscript, the generation of inner ear organoids of this cellular model is still under development and optimization. The preliminary results were presented in the Barany Society Meeting 2022 (Granada, Spain) and the Inner Ear Biology 2022 Meeting (Trieste, Italy). We have included this Conference abstracts as references 107 and 108.